# SARS-CoV-2 Morbidity in the CNS and the Aged Brain Specific Vulnerability

**DOI:** 10.3390/ijms23073782

**Published:** 2022-03-29

**Authors:** Tiziana Casoli

**Affiliations:** Center for Neurobiology of Aging, Scientific Technological Area, IRCCS INRCA, Via Birarelli 8, 60121 Ancona, Italy; t.casoli@inrca.it; Tel.: +39-071-8004-203

**Keywords:** COVID-19, SARS-CoV-2, aging, neurodegeneration, brain, glia

## Abstract

The infection by severe acute respiratory syndrome coronavirus 2 (SARS-CoV-2) can be the cause of a fatal disease known as coronavirus disease 2019 (COVID-19) affecting the lungs and other organs. Particular attention has been given to the effects of the infection on the brain due to recurring neurological symptoms associated with COVID-19, such as ischemic or hemorrhagic stroke, encephalitis and myelitis, which are far more severe in the elderly compared to younger patients. The specific vulnerability of the aged brain could derive from the impaired immune defenses, from any of the altered homeostatic mechanisms that contribute to the aging phenotype, and from particular changes in the aged brain involving neurons and glia. While neuronal modifications could contribute indirectly to the damage induced by SARS-CoV-2, glia alterations could play a more direct role, as they are involved in the immune response to viral infections. In aged patients, changes regarding glia include the accumulation of dystrophic forms, reduction of waste removal, activation of microglia and astrocytes, and immunosenescence. It is plausible to hypothesize that SARS-CoV-2 infection in the elderly may determine severe brain damage because of the frail phenotype concerning glial cells.

## 1. Introduction

In 2019, a coronavirus unknown to mankind, named severe acute respiratory syndrome coronavirus 2 (SARS-CoV-2), was first identified in the city of Wuhan in China and progressively spread all over the globe, causing, in some cases, the coronavirus disease 2019 (COVID-19), which primarily affects the respiratory system, determining bilateral pneumonia. SARS-CoV-2 is a single-stranded RNA (ssRNA) virus whose envelope is composed of a lipid bilayer and four proteins: S (spike), E (envelope), M (membrane), and N (nucleocapsid) [1,2]. To enter host cells, SARS-CoV-2 uses the S protein [3] that consists of two subunits: the S1 subunit, which binds to the host cell receptor, and the S2 subunit, which is the determinant for the fusion of virus and cell membranes. The receptor-binding domain (RBD) of S1 directly binds to the peptidase domain of the human angiotensin-converting enzyme 2 (ACE2) present on host cell membranes [4,5]. The ACE2-S1/RBD binding affinity during initial viral attachment determines the susceptibility to SARS-CoV-2 infection, and the higher transmissibility of SARS-CoV-2 over other coronaviruses is partly attributed to its greater affinity for ACE2 [6]. It has been shown that some molecular mediators can facilitate the SARS-CoV-2 infection of host cells, such as the transmembrane protease serine 2 (TMPRSS2) [7], the receptor for advanced glycation end products (RAGE) [8], the angiotensin II receptor type 2 (AGTR2) [9], the olfactory receptors [10], and neuropilin-1 (NRP1) [11], determining a great variability of SARS-CoV-2 infection capacity. ACE2 is expressed in many cell types, including neurons, astrocytes, endothelium, and smooth muscle cells of cerebral blood vessels [12]. Many viruses show tropism for the nervous tissue, causing severe neurological damage [13], and SARS-CoV-2 is not an exception, as its neurotropic properties can cause neurological disturbances. ACE2 receptors in the brain are predominantly expressed in the olfactory bulb, amygdala, hippocampus, middle temporal gyrus, posterior cingulate cortex, and brainstem [14], and the neurological symptoms of COVID-19, including hyposmia, mood disorders, cognitive impairment, sleep disorders, and dysautonomia, have been linked to dysfunctions in these brain areas [12]. Although SARS-CoV-2 can infect subjects of all ages, it can be lethal mainly for the over 80s, especially those with comorbidities, namely coronary artery disease, hypertension, obesity, and diabetes. So, it appears that the aged subjects, due to their frailty characteristics, develop a more severe clinical picture and a higher rate of systemic complications, including neurological symptoms with worse outcomes. In this review, we will try to understand the effects of SARS-CoV-2 infection on the central nervous system (CNS) and the reasons why elderly subjects are at high risk of developing critical outcomes.

## 2. Mechanisms of Neurotoxicity

### 2.1. Indirect, Direct, and Postinfectious Mechanisms

Coronaviruses are known pathogens with neuroinvasive potential and may cause neurotoxicity through indirect, direct or postinfectious mechanisms (Figure 1).

The indirect mechanisms are believed to result from the widespread alteration of homeostasis due to pulmonary, renal, hepatic, and cardiovascular injuries [15], or the use of invasive ventilation and sedation along with the side effects of drug treatments [16,17,18]. Indirect neurotoxicity may derive also from an exacerbated and dysregulated host response, called “cytokine storm”, characterized by an increased release of pro-inflammatory cytokines such as tumor necrosis factor-α (TNF-α) and interleukin-6 (IL-6), among others. If this response persists over time, it creates a state of systemic inflammation, resulting in disruption of the blood–brain barrier (BBB) with neuronal and glial damage [19,20,21].

The direct mechanisms of SARS-CoV-2 neurotoxicity derive from the straight viral diffusion inside the brain tissue. Two potential routes for coronaviruses to spread into the brain have been identified: the hematogenous pathway and the neuronal pathway. The hematogenous pathway consists of two basic mechanisms: the entry through the BBB and the transmigration of peripheral immune infected cells inside the cerebral tissue. Penetration through the BBB occurs by SARS-CoV-2 binding with ACE2 receptors present in the BBB endothelium, causing edema, intracranial hypertension and penetration of the virus into the CNS [22,23]. SARS coronaviruses can also infect bloodstream leukocytes (mainly monocytes/macrophages) and myeloid cells, which become a viral pool for the diffusion of the virus towards the CNS by a “Trojan Horse” mechanism of dissemination [24,25]. The other pathway for direct entry into the CNS is the neuronal pathway. It has been shown that some viruses can enter the nerve endings and, by a mechanism of active transport through the motor proteins kinesin, dynein and microtubules, can travel in a retrograde way, reaching the CNS through a synapse-connected route [26,27]. The olfactory nerve pathway is an excellent neuronal entrance for respiratory viruses such as SARS-CoV-2 that access the body intranasally [28].

After the acute phase, SARS-CoV-2 persistence in the CNS can lead to postinfectious mechanisms of neurotoxicity, which all stem from a misdirected host immune response and could be associated with autoimmune inflammatory and demyelinating diseases, such as various forms of encephalitis and Guillain–Barré syndrome [29]. Proposed pathogenic mechanisms include molecular mimicry between human coronaviruses and myelin basic protein and massive neuroinvasion of leukocytes and other immune cells [30]. Postinfectious CNS autoimmunity may also lead to two severe diseases: acute necrotizing encephalopathy (ANE), characterized by acute encephalopathy, seizures, reduced level of consciousness, and rapid neurological decline [31], and acute disseminated encephalomyelitis (ADEM), whose main feature is the presence of white matter lesions in the brain and spinal cord, with frequent involvement of the subcortical gray matter structures [32].

### 2.2. Viral Infections and Neurodegeneration

The link between viruses and neurodegeneration is not a novel concept. The relationship between Epstein–Barr virus (EBV), hemagglutinin type 5 and neuraminidase type 1 virus (H5N1), herpes simplex virus 1 (HSV-1) and neurodegenerative disorders has been extensively investigated [33,34,35,36,37,38,39,40] and is described in Table 1.

Among the different virus families, the Coronavirus family shows neurotropic and neuroinvasive properties in various hosts including humans [41]. Until the end of the twentieth century, only OC43 and 229E strains were known to infect humans, and they were recognized as respiratory pathogens responsible for up to 30% of common colds. Over the last 20 years, new coronaviruses infecting humans have been discovered and, among these, at least Middle East Respiratory Syndrome coronavirus (MERS-CoV), as well as SARS-CoV-1 and SARS-CoV-2, demonstrated neuroinvasive properties, proved by clinical and experimental evidence [42].

The human coronavirus OC43 (HCoV-OC43) has been shown to induce vacuolating degeneration in the gray matter of the mouse brain [43]. The observed clear round vacuoles and neuronal death are hallmarks of several neurodegenerative disorders including Alzheimer’s disease (AD) and frontotemporal dementia. These findings were again linked to the inflammatory response stimulated by the virus, although the exact cytokine profile has not been obtained.

MERS-CoV originated from bats and used camels as intermediate host for many years before jumping to humans. Clinical symptoms include high fever, cough, and dyspnea, which often leads to acute respiratory distress syndrome [44]. This virus is considered a potential neuroinvasive virus, based on clinical and experimental evidence. Studies reported that almost 28% patients with MERS-CoV developed insanity and around 9% experienced seizure attacks after infection [45,46]. Another study confirmed these trends and stated that almost 20% of patients showed neurological symptoms, such as loss of consciousness, ischemic strokes, Guillain–Barré syndrome, and paralysis [47].

SARS-CoV-1 was the etiological agent of the first SARS epidemics, which started in Asia in 2003 and spread worldwide. This virus causes the characteristic respiratory illness, which includes high fever, dry cough, and difficult breathing, and severe cases manifested by respiratory failure and death [48]. SARS-CoV-1 was found to trigger many neurological abnormalities such as encephalitis, aortic ischemic stroke, and polyneuropathy [48]. Interestingly, most SARS-CoV-1 cases displayed signs of cerebral edema and meningeal vasodilation in autopsy studies. Within the brain, the presence of SARS-CoV-1 viral particles and their RNA sequences were identified concomitantly with ischemic changes of neurons, demyelinating abnormalities, and evidence of monocyte and lymphocyte infiltrations [49].

A recent paper demonstrated that SARS-CoV-2 also could elicit changes in the CNS suggestive of a possible neurodegeneration. Yang and colleagues [50], by the analysis of 65,309 transcriptomes from frontal cortex and choroid plexus of patients with COVID-19, demonstrated that microglia and astrocyte subpopulations shared features with pathological cell states that have previously been associated with neurodegenerative diseases.

All of these findings suggest a strong correlation between viral infection and neurodegeneration. Viral infections prime the immune system to carry out powerful effects, namely through chronic inflammation, causing an enduring microglial activation and prolonged increase in inflammatory mediators. In addition, the formation of self-attacking antibodies and the initiation of mechanisms leading to specific disorders might contribute to the onset of widespread tissue damage.

### 2.3. APOE ε4 and COVID-19

The ε4 allele of APOE, the strongest genetic risk factor for AD, has been found to be linked to increased risk of infection and mortality due to COVID-19, although the biological mechanisms involved remain to be discovered [51].

Wang and colleagues [52] reported that, in cell cultures, SARS-CoV-2 infected more APOE ε4 neurons and astrocytes than their APOE ε3 counterparts. Infected neurons degenerated, while astrocytes swelled, and their nuclei broke apart. This suggests that APOE ε4 carriers may be more prone to neurological symptoms of COVID-19 than carriers of other alleles. Recent studies indicate that APOE ε4 is associated with BBB leakiness, pericyte degeneration [53], and cerebral amyloid angiopathy with capillary involvement [54]. Moreover, subjects with the APOE ε4 allele show reduced cerebral blood flow and increased subcortical ischemic white matter damage, all factors that exacerbate COVID-19 symptoms [55]. So, the association of APOE variants with blood lipids, vascular disease, and cognition should be considered to understand how APOE ε4 may increase infectivity and mortality.

## 3. Neurological Evidence of SARS-CoV-2 Damage to the CNS

With the spread of SARS-CoV-2 worldwide, many neurological symptoms have been reported [56,57], and in some cases the neurological manifestations represent the primary symptoms. While occurring in patients regardless of the severity of the disease, neurological manifestations are more frequent in already critically ill patients [58]. The main CNS complications following SARS-CoV-2 infection are impaired consciousness, encephalopathy, cranial nerve palsy, cerebral venous sinus thrombosis, ischemic stroke, intracerebral and subarachnoid hemorrhage, headache, seizure, meningitis/encephalitis, and myelitis [59]. All of these pathological events could derive from peripheral effects of SARS-CoV-2 infections, but also from direct CNS damage. To examine these possibilities, virus presence or its effects have been examined in different cerebral regions. Here, we report the latest findings on human CSF and brain autopsies.

### 3.1. CSF

In a systematic study, the main feature of COVID-19 patients with neurological complications was an increase in CSF total proteins [60]. Additionally, there are several reports of positive CSF tests for SARS-CoV-2 RNA [61]. Li and colleagues [62] analyzed 97 relevant papers and found that a total of 6.4% (30/468) of patients with COVID-19 were positive for SARS-CoV-2 in CSF, as determined by RT-PCR. Only one among the 30 patients with positive CSF for SARS-CoV-2 was analyzed for the presence of anti-SARS-CoV-2 antibodies in the CSF, and the results were positive. Among the 438 patients with negative CSF test results, 80 were tested for CSF antibodies specific for SARS-CoV-2, and 37 (46%) showed the presence of anti-SARS-CoV-2 antibodies. So, it appears that the virus is rarely present in CSF while anti-SARS-CoV-2 antibodies are more frequently found.

### 3.2. Brain Autopsies

To better understand the neuropathogenesis of COVID-19, it is important to focus on the CNS histopathological findings. One of the most frequently reported alterations is hypoxic damage with the presence of red neurons [63,64]. Other modifications include microglial activation, sometimes with nodules, and astrogliosis [65,66]. It has been postulated that reactive inflammation of microglia and astrocytes could be related to a systemic response, as in other forms of viral encephalitis [67,68]. Another hypothesis is that microgliosis and astrogliosis could be secondary to pathological mechanisms of the brain, such as infarcts or hemorrhages [69]. Indeed, in most studies, ischemic and/or hemorrhagic lesions were described, in addition to microthrombi [70].

In the previously cited review by Li and colleagues [62], results of brain autopsies in patients that died from COVID-19 were examined. SARS-CoV-2 RNA was detected in the brains of 56 out of 108 examined patients (52%), while viral proteins were found in 25 of 85 patients (29%). The analysis of 28 studies showed that CNS abnormalities were found in 134 out of 202 patients (66%). Structural modifications in the olfactory bulb/nerve were reported in 52 of 134 examined patients (39%) and consisted of microglial activation and/or lymphocytic infiltrations. Brainstem impairments such as hypoxic injury, vascular accidents, and microgliosis/lymphocytic infiltrations were reported in 78 of 134 (58%) examined patients. Cerebellar abnormalities were reported in 94 out of 134 (70%) examined patients, while cerebral structural alterations were reported in 114 of 134 (85%) patients. Both cerebellum and cerebrum abnormalities included hypoxic injury, cerebrovascular accidents, microglial activation and/or lymphocytic infiltrations. In conclusion, there is a progressive increase in brain abnormalities from olfactory structures to the cerebrum. A possible explanation of these apparent contradictions could rely on the greater vulnerability of the upper structures, such that even small viral loads might determine extensive injuries. Otherwise, cerebellum and cerebrum alterations, given the patient age, could have been already present in the brain, even before SARS-CoV-2 infection. It should also be considered that the viral detection methods or sampling procedures may not be suitable or sufficient to reveal the CNS infection and that the pathological changes could have been determined by factors other than infection. Another concern about autopsy studies is that molecules and cells of blood origin can be found in the brain parenchyma due to the leakiness of brain arteries and veins in COVID-19 patients. During the neuropathological investigation, it would be difficult to distinguish them from residential molecules and cells, rendering the interpretation of the autoptic samples quite complex.

## 4. COVID-19 and the Aged Brain

### 4.1. Loss of Homeostatic Control in the Elderly

The infection by SARS-CoV-2 leads to severe COVID-19 mostly in elderly subjects and frail individuals affected by pre-existing chronic pathologies. The reason why it happens could be found in the impaired immune defenses of these patients and/or in any of the altered homeostatic mechanisms that contribute to the aging phenotype [71,72]. These changes occur at the biological level over time in living organisms, and those with a primary role are genomic instability, telomere shortening, cellular senescence, epigenetic changes, and mitochondrial impairment [73].

The accumulation of DNA somatic mutations due to genomic instability affects the fidelity of proteins and the regulation of gene expression. Spontaneous somatic mutations have been demonstrated in human B lymphocytes, and it has been suggested that they may contribute to functional decline of these cells in the elderly [74].

Short telomeres have been hypothesized to contribute to the aging process [75]. Indeed, several reports indicate that short telomeres may be associated with increased risk of cardiovascular events [76], reduced immune response [77], and mortality [78,79].

Senescent cells accumulate in tissues during aging and could develop the “senescence-associated secretory phenotype” (SASP) that determines secretion of proinflammatory cytokines, chemokines, growth factors, and matrix proteases [80]. It has been proposed that the accumulation of senescent cells and the negative effects of SASP proteins on intercellular matrix and on progenitor cells cause tissue degeneration and chronic inflammation [81].

One of the most studied epigenetic changes in aging and age-related chronic diseases is DNA methylation. The percentage of methylated sites along the DNA sequence can be used to define an “epigenetic clock” that identifies individuals who are biologically older or younger than their chronological age [82,83]. Consistent with this notion, “epigenetically older” individuals have a higher risk of developing several age-related diseases and premature mortality from all causes [84].

The accumulation of damage in mitochondria and mitochondrial DNA (mtDNA) induces important changes in cellular physiology by reducing energy availability and increasing production of reactive oxygen species (ROS) that damage macromolecules [85]. Recent data support the hypothesis that mtDNA copy number and degree of heteroplasmy, assessed in human blood and in tissue biopsies, provide information on mitochondrial functioning that is relevant for aging and age-related diseases [86,87,88].

The described age-dependent changes modify cellular metabolism in every apparatus, including the CNS, and predispose individuals to be vulnerable towards the infection by SARS-CoV-2 (Table 2). Hallmarks of aging have been shown to be all interconnected, thus targeting any single hallmark could result in beneficial changes of the others [89].

### 4.2. Aged Brain Abnormalities Predisposing to Neurological Manifestations of COVID-19

Analysis of the brain at the cellular and molecular levels shows that many of the biomarkers of aging, evident in other tissues, are present in the brain as well. In addition, brain displays some specific changes that depend on its peculiar characteristics and can involve neurons and glia. The main changes of neurons include impaired lysosomal and proteasomal degradation, altered calcium regulation, and distorted adaptive stress response.

Evidence that lysosomal and proteasomal degradation is impaired comes from studies showing age-related intracellular accumulation of autophagosomes, dysfunctional mitochondria, and polyubiquitinated proteins [90,91].

Neurons from aged brain also display a remarkable dysregulation of Ca^2+^ lowering after its synaptic activity-dependent influx. The aberrant elevation of cytoplasmic Ca^2+^ levels has widespread effects that result in the dysregulation of protein phosphorylation, cytoskeletal dynamics, and gene expression [92,93], leading in some cases to neuronal cell death [94].

The three major initiators of adaptive cellular stress responses are ATP consumption, Ca^2+^, and ROS, which can induce the expression of genes encoding proteins that mitigate cellular stress and eliminate or repair damaged molecules [95]. The adaptive stress response of neurons may become impaired during aging, thereby rendering neurons vulnerable to injury [96].

Although these neuronal changes could contribute indirectly to CNS damage induced by SARS-CoV-2, glial cell alterations could play a more direct role, as they are involved in CNS immune defense and response to viral infections (Table 2).

Aging is associated with the accumulation of dystrophic forms of glial cells showing a reduced ability to mount a gliotic response, hence facilitating the brain damage caused by imported pathogens [105]. Furthermore, the waste collection activity declines with age, thus affecting the clearance of numerous by-products and toxic substances [106]. These glial abnormalities make the brain more vulnerable to systemic pathologies and damage from infections.

In addition, the aged brain is characterized by a general condition of low-grade inflammation, the so-called neuroinflammation, which is involved in neurodegenerative and neuropsychiatric disorders. Neuroinflammation is mainly regulated by glial cells, such as microglia and astrocytes. During neuroinflammation, microglia become activated, and consequently, the production of chemokines and cytokines such as TNF-α, IL-6, interleukin 1ß (IL-1β), interferon- γ (IFN-γ) and chemokine (C-C motif) ligand 2 (CCL2) is significantly increased [107]. The resulting oxidative stress further activates neighboring microglial cells, thus causing a chronic activation [108]. The presence of neuroinflammation with age is also indicated by an increase in GFAP positive astrocytes [97,98]. Neuroinflammation interferes with brain function and may cause structural damage, influence regeneration, modulate synaptic remodeling, and induce neuronal cell death.

Since neuroinflammation can be induced or worsened by the SARS-CoV-2 infection itself, the role of neuroinflammatory mechanisms could be central in a vicious circle leading to an increase in hyperproduction of cytokines and mortality risk in aged COVID-19 patients.

During aging, glial cells also show the signs of immunosenescence [99]. Immunosenescence is most often described as an age-related change to the adaptive immune system that results in impaired ability to fight infections, in blunted response to new antigens, and in increased incidence of autoimmunity, contributing to the development of aging-associated diseases [100]. A constellation of maladaptive aging-related immune system changes occurs in immunosenescence, such as the limited ability of CD4+ and CD8+ T-cell subtypes to mount robust responses to new epitopes and the decreased pool of naive CD4+ and CD8+ T cells [101]. Previous studies have shown that telomere shortening occurs in rat microglia in vitro [102] and in rat cerebellum and cortex with age in vivo [103]. Furthermore, astrocytes undergo changes that indicate the presence of the SASP, such as: (i) increased levels of intermediate GFAP and vimentin filaments, (ii) increased expression of several cytokines, e.g., IL-6, IL-1ß, TNF-α, and high mobility group box 1 (HMGB1), and (iii) age-related ultrastructural changes in nuclei and accumulation of lipofuscin in cytoplasm [104].

To sum up, it is plausible to hypothesize that SARS-CoV-2 infection in the elderly, mostly in conditions of neuroinvasivity, may determine severe damages to the brain because nervous tissue is already in a compromised state and cannot properly face the injuries caused by the virus (Figure 2). So, the treatment of age-related anomalies could be useful not only to prevent diseases, but also to reduce COVID-19 detrimental effects.

## 5. Neurological Consequences, Cognitive Dysfunction and Possible Mechanisms of “Long COVID”

The long-term neurological consequences of COVID-19 are currently unknown; however, many reports describe persistent symptoms even months after resolution of the infection, including impaired smell and/or taste, chronic fatigue, and impaired cognition [59]. Long-term complications of infection are referred to as “Long COVID”, and it is not clear if SARS-CoV-2 is completely eradicated from possible brain sites of infection after recovery from COVID-19. Although very limited evidence exists to date on the pathophysiological mechanisms implicated in the manifestation of “Long-COVID”, neuroinflammatory, prothrombotic, hypoxic, metabolic, and apoptotic cascades, activated during acute COVID-19, are thought to persist even in the absence of an evident infection, affecting the brain’s normal function [109]. It has been demonstrated in patients with “Long COVID” that some brain areas, such as the brainstem and cerebellum, show a reduced glucose metabolism possibly attributed to oxidative stress, mitochondrial dysfunction, and neuronal degeneration [110]. A single case report showed that a 78-year-old woman who recovered from COVID-19 and had three consecutive SARS-CoV-2 PCR negative nasopharyngeal swabs, died of a sudden cardiac arrest, and postmortem investigation revealed residual virus in the lungs [111]. These findings suggest that the virus may not be definitively cleared in some patients that appear to have recovered and raises the possibility that SARS-CoV-2 may evade immune surveillance, at least to some degree. If the virus persists in the brain after recovery, some long-term consequences could be envisaged impacting cell function and homeostasis. Varatharaj and colleagues [112] reported that “altered mental status” was listed as one of the clinical syndromes associated with COVID-19 and defined it as an “acute alteration in personality, behavior, cognition, or consciousness”. In the same study, 31% of the patients analyzed had an altered mental status following COVID-19, and nearly 5% had dementia-like cognitive symptoms. It should be noted that manifestations of cognitive impairment are more prevalent in older individuals and in those with severe infections [113]. Lu et al. [114] recorded data of 60 patients during acute SARS-CoV-2 infection and at a 3-month follow-up. The proportion of patients with memory loss more than doubled from 13.3% during the acute disease to 28.3% at the follow-up, demonstrating a long-term impact of COVID-19 on cognition. As regards the possible mechanisms of cognitive impairment after COVID-19 recovery, a study of 310 patients showed that those with new neurological symptoms after healing had higher levels of t-tau, neurofilament light chain (NfL), GFAP, and p-tau-181 in their blood, as well as indicators of inflammation such as C-reactive protein, compared to patients without neurological symptoms after healing, suggesting that former patients could have an underlying Alzheimer-related pathology [115]. Another important effect of SARS-CoV-2 infection in the brain could be the diversion of the bioenergetics of infected cells from the normal neuronal metabolism to the support of viral replication. It is possible that mitochondrial targeting by the virus may be the substrate for the emergence of cognitive impairment or ‘brain fog’ [116]. Recent research has shown that redirecting mitochondrial activity also occurs during Ebola, Zika, and influenza A virus infections [117]. Viral targeting of mitochondria may be an evolutionarily conserved adaptive process for viruses and bacteria, as mitochondria are organelles with a prokaryotic origin [118]. Cognitive impairments in patients recovered from COVID-19 might be linked to inflammatory markers. Zhou and colleagues, investigating the impacts of COVID-19 on cognitive functions in patients who recovered from the viral infection, found that there was a correlation of cognitive impairment, as measured by the continuous performance test (CPT), with levels of C-reactive protein [119]. They concluded that the cognitive impairment in patients recovered from COVID-19 might be linked to the persisting inflammation.

## 6. Concluding Remarks

Although the neuropathological changes induced in the brain by SARS-CoV-2 infection are not different in adult and elderly subjects, the overall effects are significantly worse in aged subjects. Our hypothesis is that the aging frail phenotype contributes mainly to this outcome, especially the one involving glial cells, as they are involved in the immune defenses and in the response to viral infections. Immunosenescence and the persistent low grade of neuroinflammation are additional critical points weaking the reactivity to SARS-CoV-2 attack. The observation of cognitive impairment following recovery from COVID-19, especially in older individuals, strengthens the link between the viral infection and neurodegeneration induced by the virus. The use of a geroscience approach could then be a productive way not only to prevent the age-related pathologies, but also to tackle possibly recurrent pandemics.

## Figures and Tables

**Figure 1 ijms-23-03782-f001:**
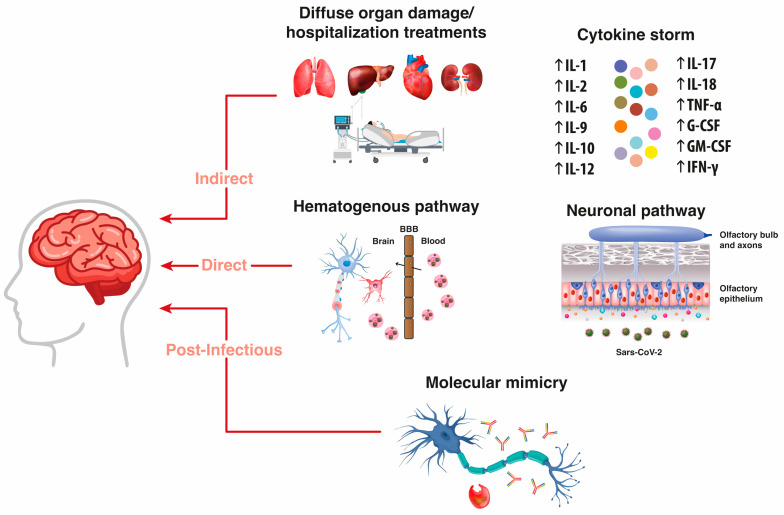
Indirect, direct and postinfectious mechanisms of neurotoxicity due to SARS-CoV-2 infection. The indirect mechanisms result from pulmonary, hepatic, cardiovascular, and renal injuries, as well as from the side effects of hospitalization treatments. Indirect neurotoxicity may derive also from an abnormal host response called “cytokine storm”, characterized by an increased release of pro-inflammatory cytokines. The direct mechanisms of SARS-CoV-2 neurotoxicity originate from the straight viral diffusion into the brain through the hematogenous pathway and/or the neuronal pathway. The hematogenous pathway consists of the entry of peripheral immune infected cells into the brain by the blood–brain barrier (BBB). In the neuronal pathway, viruses reach the nerve endings and, by a mechanism of retrograde active transport, reach the central nervous system by a synapse-connected route. Here, the entry through the olfactory epithelium and olfactory nerve is shown. Post-infectious mechanisms stem mainly from molecular mimicry between human coronaviruses and myelin basic protein (antibodies attacking myelin sheath and a phagocytosing macrophage are represented).

**Figure 2 ijms-23-03782-f002:**
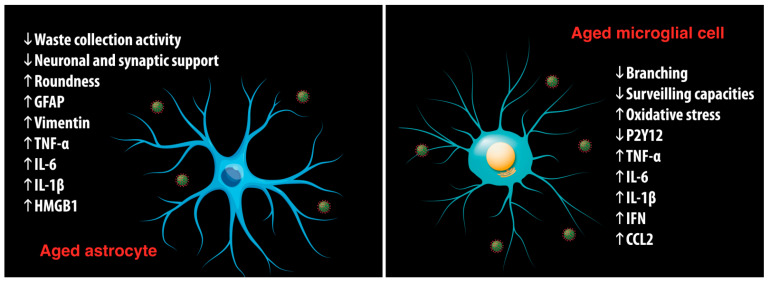
Age-associated changes of astrocytes and microglia amplifying the negative effects of SARS-CoV-2. Aged astrocytes undergo functional changes such as decreased waste collection activity and neuronal support. In addition, they show increased levels of glial fibrillar acidic protein (GFAP) and vimentin filaments as well as increased expression of several cytokines and high mobility group box 1 protein (HMGB1). Aged microglia show decreased branching and surveilling capacities as well as increased oxidative stress. The production of chemokines and cytokines such as TNF-α, IL-6, IL-1β, IFN and CCL2 is significantly increased. All of these changes determine a specific vulnerability of the aged brain to SARS-CoV-2 infection.

**Table 1 ijms-23-03782-t001:** Viral infections and neurodegeneration.

Virus Strain	Effect of Infection Triggering Neurodegeneration	References
Epstein–Barr	Brain tissue damage initiated by specific response of CD8+ T cell to infection	[33,34]
	Autoimmunity by molecular mimicry with myelin antigens	[35]
Hemagglutinin type 5 and neuraminidase type 1	Increased levels of interleukin-18, interleukin-6, granulocyte colony-stimulating factor, and monocyte chemoattractant protein-1	[36]
	Microglial activation and dopaminergic neuronal loss in the substantia nigra	[37]
Herpes simplex 1	Neurotoxic amyloid-β accumulation	[38,39]
	Tau phosphorylation	[40]

**Table 2 ijms-23-03782-t002:** Determinants of vulnerability in the aged brain.

Age-Associated Changes in the Brain	References
**General**	
Genomic instability	[74]
Telomere shortening	[75,76,77,78,79]
Cellular senescence	[80,81]
Epigenetic changes	[82,83,84]
Mitochondrial impairment	[85,86,87,88]
**Specific/neurons**	
Impaired lysosomal and proteasomal degradation	[90,91]
Altered calcium regulation	[92,93,94]
Distorted adaptive stress response	[95,96]
**Specific/glia**	
Increased activation of microglia and astrocytes	[97,98,99,100]
Immunosenescence	[101,102,103,104,105,106]

## Data Availability

Not applicable.

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
