# Peer review of "SARS-CoV-2 Morbidity in the CNS and the Aged Brain Specific Vulnerability"

_ijms, 2022, doi:10.3390/ijms23073782_

Round 1

Reviewer 1 Report

Manuscript Number: 1629775

Title: Sars-CoV-2 morbidity in the CNS and the aged brain-specific vulnerability

In this review paper, the authors discuss Sars-CoV-2 associated morbidity in the CNS and the vulnerability of the aged brain. The paper is well written, the abstract and the introduction parts are ok. However, the paper needs some modification so that it will be suitable for publication.

  1. In section 2 the mechanism of neurotoxicity, the authors could mention in detail the indirect entry of Sars-CoV-2 in the brain (routes of infection). The direct enter pathways are adequately described.
  2. figure 1 on page 3 needs modification. The illustration regarding the Hematogenous pathway should rotate 180 degrees so that the brain will be on the left side and blood to the right. Of course, the arrow has to point from blood to brain. The illustration of the Neuronal pathway needs labeling (the authors need to show which one is the viruses, which one is the epithelial surface and nerve endings. the illustration about Neuroinvasion by immune cells is better to remove because it is part of the hematogenous pathway.
  3. section 2.2 viral infection and neurodegeneration. in this part, the authors should focus on the role of Coronavirus families in neurodegeneration. There is enough literature and animal models which could be mentioned here. It is better to remove the part which explains the role of Herpesviridae families in neurodegeneration because it is well characterized and it does not bring new information.
  4. Brain autopsies interpretation.

We know that SAR-CoV-2 infection causes brain arteries and veins to become thin and leaky and the autopsies of people who have died from COVID-19 show leaky blood in different areas of the brain that allow molecules and blood cells that are normally excluded from the brain to move from the bloodstream into the brain, therefore the interpretation of the autopsies samples is difficult. The authors at least add such concern in this section.

  1. in the section Neurological consequences, cognitive dysfunction, and possible mechanism of long covid. Since there is no persistent infection and no viral protein expression, how is this long-term effect of COVID-19 infection explained? Do you think SAR-COV-2 can undergo a latency period in CNS?

    Manuscript Number: 1629775

    Title: Sars-CoV-2 morbidity in the CNS and the aged brain-specific vulnerability

    In this review paper, the authors discuss Sars-CoV-2 associated morbidity in the CNS and the vulnerability of the aged brain. The paper is well written, the abstract and the introduction parts are ok. However, the paper needs some modification so that it will be suitable for publication.

    1. In section 2 the mechanism of neurotoxicity, the authors could mention in detail the indirect entry of Sars-CoV-2 in the brain (routes of infection). The direct enter pathways are adequately described.
    2. figure 1 on page 3 needs modification. The illustration regarding the Hematogenous pathway should rotate 180 degrees so that the brain will be on the left side and blood to the right. Of course, the arrow has to point from blood to brain. The illustration of the Neuronal pathway needs labeling (the authors need to show which one is the viruses, which one is the epithelial surface and nerve endings. the illustration about Neuroinvasion by immune cells is better to remove because it is part of the hematogenous pathway.
    3. section 2.2 viral infection and neurodegeneration. in this part, the authors should focus on the role of Coronavirus families in neurodegeneration. There is enough literature and animal models which could be mentioned here. It is better to remove the part which explains the role of Herpesviridae families in neurodegeneration because it is well characterized and it does not bring new information.
    4. Brain autopsies interpretation.

    We know that SAR-CoV-2 infection causes brain arteries and veins to become thin and leaky and the autopsies of people who have died from COVID-19 show leaky blood in different areas of the brain that allow molecules and blood cells that are normally excluded from the brain to move from the bloodstream into the brain, therefore the interpretation of the autopsies samples is difficult. The authors at least add such concern in this section.

    1. in the section Neurological consequences, cognitive dysfunction, and possible mechanism of long covid. Since there is no persistent infection and no viral protein expression, how is this long-term effect of COVID-19 infection explained? Do you think SAR-COV-2 can undergo a latency period in CNS?

Reviewer 2 Report

This is the review manuscript on the Sars-CoV-2 morbidity in the CNS and the aged brain-specific vulnerability. The manuscript is well-written and informative.  But I require the minor changes as written below.

  1. The author described that ACE2 is expressed in many cell types(page 1 line 42). Can the author show the specific cells in the brain? Does this expression affect the pathogenesis of brain damage?
  2. SARS-CoV-2 or SARs-Cov-2 or Sars-CoV-2? Please integrate them into one expression.
  3. Explain more specifically on "structural abnormality" (page 6 line199)
